# Clinical Significance of Ventricular Premature Contraction Provoked by the Treadmill Test

**DOI:** 10.3390/medicina58040556

**Published:** 2022-04-18

**Authors:** Min-Jung Bak, Hee-Jin Kwon, Ji-Hoon Choi, Seung-Jung Park, June-Soo Kim, Young-Keun On, Kyoung-Min Park

**Affiliations:** Samsung Medical Center, Division of Cardiology, Heart Vascular and Stroke Institute, Sungkyunkwan University School of Medicine, Seoul 06351, Korea; cindy629@naver.com (M.-J.B.); subakhj@gmail.com (H.-J.K.); jihoon.choi@samsung.com (J.-H.C.); seungjung.park@samsung.com (S.-J.P.); js58.kim@samsung.com (J.-S.K.); yk.on@samsung.com (Y.-K.O.)

**Keywords:** treadmill test, ventricular premature beats, coronary revascularization, exercise phase, recovery phase

## Abstract

*Background and Objectives*: The treadmill test (TMT) is a predictive tool for myocardial ischemia. Recently, exercise-provoked ventricular premature contracture (VPC) during TMT was shown to have a relation with coronary artery disease and cardiovascular mortality. Therefore, we evaluate clinical correlates of exercise-provoked VPC and compare the predictive power for myocardial ischemia and cardiovascular events. Method: Data of 408 patients (≥18 years of age) who underwent TMT for work up of angina, palpitation, dyspnea, syncope, or arrhythmia between February 2015, and January 2016, were collected with consent at Samsung Medical Center, Seoul, Republic of Korea. Among total of 408 patients, 208 were excluded according to the previous history of PCI or CABG, previous MI, decreased left ventricular ejection fraction lower than 50%, arrhythmia that could affect ST-segment change on ECG. Results: Among 200 patients, 32 (16.0%) developed exercise-provoked VPC (21 patients in the exercise phase, 20 patients in the recovery phase). Of them, 20 patients (10.0%) showed positive TMT, and 4 patients (2.0%) underwent revascularization after TMT. Among 21 patients showing exercise phase VPC, 5 (23.8%) showed positive TMT results. In patients younger than 65 years, exercise phase VPC was associated with positive TMT (odds ratio 6.879, 1.458–32.453) considering demographics and previous medical history in multivariable analysis. Among the 20 patients showing recovery phase VPC, 2 (10.0%) underwent revascularization after TMT. In multivariable analysis, recovery phase VPC was associated with revascularization (odds ratio 9.381, 1.144–76.948) considering age, sex, BMI, and TMT result. *Conclusion*: VPC during the treadmill test was a useful predictor of myocardial ischemia in this study.

## 1. Introduction

Change in ST-segment in the treadmill test (TMT) [1,2,3,4] is traditionally used to predict myocardial ischemia. In addition to the ST depression sign during TMT, studies have been conducted on whether exercise-provoked arrhythmia, a finding was frequently seen during TMT, could predict myocardial ischemia or mortality, VPC can occur by ectopic nodal automaticity, re-entrance signaling, or triggered beat. Myocardial ischemia is known to reduce the arrhythmic threshold and ectopic pacemaker cells can carry a sub-threshold potential for firing [5,6]. Some studies have suggested exercise-induced ventricular arrhythmia to be associated with exercise-induced ischemia [7,8], and other studies have reported opposing results [9]. There are reports [10,11,12,13] that patients with ventricular activity noted during exercise have higher mortality compared to those with no such activity. However, other reports conclude no such relationship with mortality [14,15] or coronary artery disease [16,17,18]. As such, there have been many reports that have studied the relationship between exercise-provoked VPC and cardiovascular events. However, most of those studies were conducted in Europe or the USA, and there is a paucity of data on Asian populations. Therefore, we evaluate clinical correlates of exercise-provoked VPC and compare the predictive power for myocardial ischemia and cardiovascular events in the Asian population.

## 2. Materials and Methods

### 2.1. Study Population

Data of 408 patients (≥18 years of age) who underwent TMT for work up of angina, palpitation, dyspnea, syncope, or arrhythmia between February 2015, and January 2016, were collected with consent at Samsung Medical Center, Seoul, Republic of Korea. This study complied with the Declaration of Helsinki, and the research protocol was approved by the Ethics committee of Samsung Medical Center. Of these patients, 51% were referred for testing because of angina pectoris; 39% for non-ischemic chest pain, resting ECG abnormality, or elevated risk factors; 5% to evaluate arrhythmias; 3% for exercise capacity assessment; less than 2% for dyspnea on exertion; and the rest for miscellaneous reasons.

### 2.2. Data Collection

All exercise tests were supervised, and all results were read by two of the investigators (MJP and JHC). A thorough clinical history including medications and risk factors was recorded prospectively at the time of the exercise test using computerized forms.

### 2.3. Exercise Test

The treadmill test was conducted according to a symptom-limited Bruce protocol that divides step one to increase exercise capacity slowly. Exercise capacity was estimated in metabolic equivalents based on treadmill speed and grade, and 12-lead ECG data were recorded with Quinton Q-stress. An abnormal ST-segment response was defined as 1 mm or more of horizontal or down-sloping depression measured visually at the J junction. Underlying medical history, family history, follow-up data, and death was collected from hospital medical records. Exclusion criteria were previous history of coronary revascularization (percutaneous coronary intervention or coronary artery bypass graft), previous MI, left ventricular ejection fraction lower than 50%, arrhythmia during TMT, and any disease that could affect ST-segment change on ECG. A cardiovascular event was defined as a history of documented coronary artery disease on coronary angiography or computed tomography angiography, myocardial infarction (MI), stroke, percutaneous coronary intervention (PCI), or coronary artery bypass graft surgery (CABG).

### 2.4. Classification of VPC

Exercise-provoked VPC was noted if frequent VPCs (more than 10% of all VPCs during any 30-s ECG recording) or ventricular tachycardia (three or more consecutive VPCs) were visually detected during the exercise test or recovery. Resting VPC was recorded if a VPC was detected in the 10-s ECG prior to exercise. We divided the patients into the VPC group (VPCY) and the non-VPC group (VPCN) based on the results of the TMT. We defined VPCY as VPC that occurred during TMT (exercise phase or recovery phase) regardless of evidence of resting phase VPC. Further, we compared the clinical outcomes of patients with VPC according to phase, exercise (VPCE), or recovery (VPCR). Categorization of VPCE means that VPC occurred during the exercise phase of TMT, while VPCR is VPC that occurred during the recovery phase. There was an overlap between the VPCE and VPCR groups.

### 2.5. Statistics

Continuous variables were tested using an independent two-sample t-test and are presented as mean ± standard deviation. Categorical data were compared using the Chi-square test and are presented as numbers and relative frequency (%). The multivariable models comprised the covariates that were significant on univariate analysis (*p* < 0.1) or that were clinically relevant. All probability values were two-sided, and a *p*-value < 0.05 was considered statistically significant. Statistical analyses were performed using SPSS statistics 20 (SPSS Inc., Chicago, IL, USA).

## 3. Results

Among total of 408 patients, 208 were excluded according to the previous history of PCI or CABG, previous MI, decreased left ventricular ejection fraction lower than 50%, arrhythmia like paroxysmal supraventricular ventricular tachycardia during TMT, and any disease that could affect ST-segment change on ECG, such as atrial fibrillation/atrial flutter, bundle branch block, or Wolff-Parkinson-White syndrome. Finally, we enrolled 200 patients with a median follow-up of 32 months. Among these 200 patients, 32 patients (16.0%) developed VPC. Of these 32 patients, 21 were exercise VPC, 20 were recovery VPC and 11 were both exercise and recovery phase VPC (Figure 1).

In baseline characteristics of demographics, past medical history, clinical finding, and exercise test record, there was no significant deference between VPCY and VPCN; between VPCE and all others; or between VPCR group and all others. The VPCR group had greater family history of sudden death compared to the others (*p* = 0.014), while the VPCE group showed higher prevalence of positive TMT result (*p* = 0.042). (Table 1) Based on baseline characteristics, we analyzed clinical correlation with logistic regression in the VPCY, VPCE, and VPCR groups. The VPCY group (OR 5.786 (1.111–30.120)) and VPCR group (OR 10.875 (2.026–58.363)) showed significant correlation with family history of sudden cardiac death (Appendix A).

Among 200 enrolled patients, 20 (10%) showed positive TMT (TMT (+)). There was no significant difference in incidence of VPCY between TMT (+) and TMT (−) patients (25% vs. 15%, *p* = 0.329). In the study, 8 patients (4%) were diagnosed with significant coronary artery disease in coronary angiography. Further, 4 patients (2%) underwent revascularization. Among the TMT (+) patients, 5 (25%) were diagnosed with significant coronary artery disease in coronary angiography, a significantly higher percentage compared to the 1.7% in TMT (−) patients (*p*-value < 0.001). Among 5 patients, one (5.0%) underwent PCI and one (5.0%) underwent CABG during follow-up. The trend of revascularization rate was higher in patients with VPCY than with VPCN in TMT (+) patients (VPCY vs. VPCN: 20.0% vs. 6.7%, *p* = 0.447), but the difference was not significant.

To determine the relationship of VPC with TMT (+), we conducted univariable and multivariable logistic regression. In univariable logistic regression, VPCY and TMT (+) had no significant relation (OR = 1.889 (0.634–5.628)), *p*-value = 0.254). After dividing VPCY into specific phases, exercise phase VPC (OR = 3.417 (1.098–10.628), *p*-value = 0.034) showed significant relation with TMT (+), while recovery phase VPC (OR = 1.692 (0.450–6.367), *p* value = 0.437) did not. In multivariable analysis, VPCE had a relation with positive TMT in patients younger than 65 years (OR = 6.879 (1.458–32.453), *p* value = 0.015), after correcting for demographics and previous medical history (Table 2). In the prediction of coronary artery disease which is significant stenosis in more than 1 vessel confirmed by coronary angiography, age (OR = 1.157 (1.030–1.300), *p* value = 0.014) and positive TMT result (OR = 22.224 (3.267–151.154), *p* value = 0.002) was significantly related after correcting for sex, BMI and the presence of exercise phase VPC or recovery phase VPC. In the prediction of revascularization, VPCR had a relation with revascularization (OR = 9.381 (1.144–76.948), *p*-value = 0.037) after correcting for age, sex, BMI and TMT result(Table 3). When we compared baseline characteristics under the assumption that VPCE and VPCR groups were independent without overlap, there was no difference in demographics, previous medical history, or exercise test results between the two groups (Appendix A).

## 4. Discussion

With this study, we suggested that exercise-provoked VPC could be the sign of myocardial ischemia with respect to predicting revascularization events after TMT even in the Asian population.

With respect to revascularization, VPCR showed significant predictive value in our study. In the previous Framingham Heart Study [12], there was no relation between coronary heart disease (CHD) event and exercise-provoked VPC. However, another study [19] conducted on patients with suspected coronary artery disease showed a significant relation between CHD events and exercise-induced ventricular arrhythmia. Considering the results of those previous studies, the reason that exercise-provoked VPC and revascularization were found to be related in the present study seems to have been influenced by the analysis of symptomatic patients with suspected coronary artery disease, not the community population. Among exercise-induced VPCs, recovery phase VPCs are known to predict myocardial ischemia better than exercise-phase VPCs, which is also consistent with the results of our study [19,20]. This means that VPCR was closer to achieving actual clinical relevance than was VPCE. Another notable point in this study is that recovery phase VPC was associated with revascularization independently of TMT results. Therefore, even if the TMT result is negative, paying attention to the recovery phase VPC will be helpful in finding patients with significant epicardial coronary artery stenosis.

In our study, exercise phase VPC showed a relation with positive segments-segment change in TMT after adjusting for cardiovascular risk factors (age, sex, BMI, DM, HTN, dyslipidemia, history of CVA or PAD). Unlike the recovery phase VPC, the exercise phase VPC was not associated with epicardial coronary artery stenosis and clinical outcome but was correlated with ST depression. ST depression in ECG is also known to be associated with coronary microvascular dysfunction (CMD) in the absence of epicardial coronary artery stenosis [21]. However, in this study, since neither instantaneous flow ratio (IMR) nor coronary flow reserve (CFR) was measured during coronary angiography, the presence of CMD could not be confirmed. Based on these results, future studies are expected to verify the predictability of exercise phase VPC with coronary microvascular disease. The analysis of the relationship between exercise provoked VPC and TMT results was established in patients under 65 years of age in this study. Considering that TMT is mainly performed in ambulatory young patients, this restriction does not seem to have a significant impact on the interpretation of the results. Although there are reports that exercise-provoked VPC increased with age [9,10,12,22] and male sex [12], there was no such difference in incidence in the present study. Here, patients with resting phase VPC showed more frequent exercise-provoked VPC, in agreement with a previous study [22].

Exercise-provoked VPC patients with or without known coronary artery disease had a more frequent family history of sudden cardiac death, corresponding to a previous study [23]. This indicates that such patients could have unknown familial heart failure with preserved ejection fraction [24,25]. We excluded patients according to baseline ejection fraction. If we had excluded patients based on echocardiographic morphology, such as hypertrophy, valve abnormality, and regional wall motion abnormality, different results may have been observed.

There is no clear pathophysiology of exercise provoked VPC. Some studies have suggested that exercise-induced ventricular arrhythmia is associated with exercise-induced ischemia [7,8], though other studies with a myocardial infarction group or healthy population have reported opposing results [9,12]. It is possible that exercise-provoked VPC is caused by viable ventricular ischemic cells with altered electrical conductance.

The treadmill test is a useful modality in the outpatient clinic because it is simple, easy to interpret, shows the low inter-observer difference, and is inexpensive and noninvasive. Although the convenience and accessibility of TMT are good advantages, low negative predictive value and low positive predictive value are the limitations of TMT, especially compared with updated tools such as stress echocardiography or perfusion cardiovascular magnetic resonance imaging [3,26,27,28]. Exercise provoked VPC during TMT could compensate for this limitation of the TMT. There is much evidence that exercise-provoked VPC is related to cardiovascular mortality [10,11], but the predictive value combined with standard ST change criteria has not been established. There was no statistical difference in VPCY incidence based on TMT results in our study. If ST change and exercise-provoked VPC have different mechanisms, complementary effects would be expected for TMT.

### Limitations

The present study population comprised patients visiting a hospital for specific symptoms. Because of this, they were relatively old and showed a higher percentage of VPC compared to another study of healthy asymptomatic patients [11]. Our results are more representative of those of a general ambulatory clinic. We defined exercise-induced VPC during TMT as any single VPC during the exercise or recovery phase with or without evidence of VPC on resting ECG. Thus, VPCY could overlap with resting phase VPC. However, based only on the 10-s baseline ECG, we cannot exclude resting phase VPC. Selection bias could have been produced by the required consent of the study population and by retrospective analysis of prospectively collected data. Although there also is a limitation as a single-center study, the residence of patients spread nationwide rather than being limited to Seoul because of center characteristics. We collected data from a large population over a short period (1 year), follow up period (32 months) was relatively short, and follow-up data were collected from in-hospital medical records. There were only three mortality cases without cardiovascular cause during follow-up, though there is a possibility of underestimated mortality cases. If we trace patients consistently and collect data based on in-hospital medical records and on the National Institute of Health, a reanalysis would produce a stronger conclusion.

## 5. Conclusions

Since the exercise phase, VPC is related to ST depression, and the recovery phase VPC could predict revascularization, exercise-provoked VPC during TMT can be used as a marker of myocardial ischemia.

## 6. Patents

This section is not mandatory but may be added if there are patents resulting from the work reported in this manuscript.

## Figures and Tables

**Figure 1 medicina-58-00556-f001:**
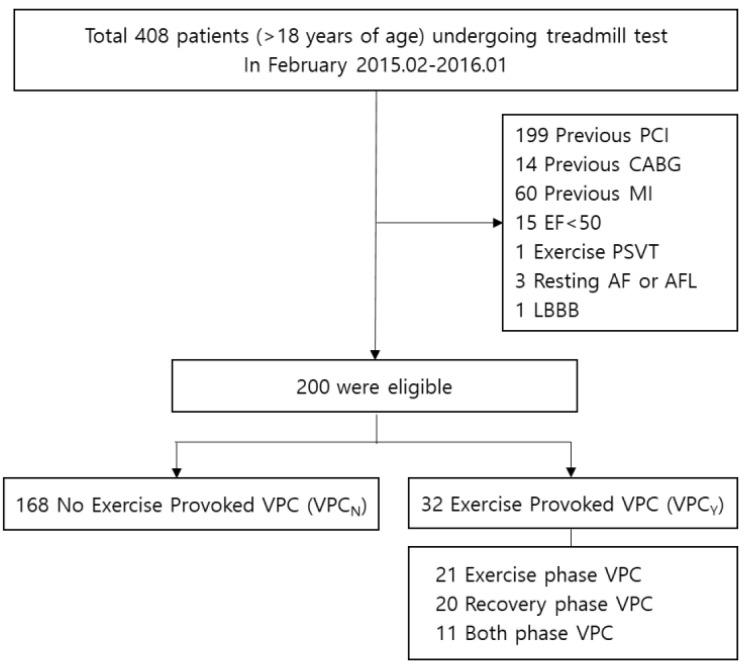
Study flow chart with number of patients.

**Table 1 medicina-58-00556-t001:** Baseline characteristics (*n* = 200).

	VPC_Y_ (*n* = 32)	VPC_N_ (*n* = 168)	*p* Value (VPC_Y_ vs. VPC_N_)	VPC_E+_ (*n* = 21)	*p* Value (VPC_E_ vs. Others)	VPC_R_ (*n* = 20)	*p* Value (VPC_R_ vs. Others)
**Demographics**							
**Age**	56.4 ± 16.6	52.7 ± 14.0	0.180	57.7 ± 16.1	0.139	53.8 ± 17.9	0.869
**Sex (male)**	27 (84.4%)	122 (72.6%)	0.162	3 (14.3%)	0.213	5 (25.0%)	0.957
**Height**	167.0 ± 7.7	167.0 ± 8.7	0.988	166.3 ± 8.0	0.696	165.7 ± 7.2	0.495
**Weight**	68.4 ± 9.4	68.7 ± 11.9	0.888	69.8 ± 10.2	0.649	67.3 ± 10.1	0.586
**BMI**	24.5 ± 2.8	24.6 ± 3.1	0.951	25.2 ± 2.8	0.319	24.5 ± 2.7	0.907
**Family Hx**							
**FHxCAD**	4 (12.5%)	14 (8.3%)	0.498	4 (19.0%)	0.103	3 (15.0%)	0.399
**FHxSCD**	3 (9.4%)	3 (1.8%)	0.053	1 (4.8%)	0.491	3 (15.0%)	0.014
**Pre. Medical Hx**							
**DM**	5 (15.6%)	23 (13.7%)	0.782	3 (14.3%)	1.000	3 (15.0%)	1.000
**HTN**	12 (37.5%)	71 (42.3%)	0.616	9 (42.9%)	0.894	9 (45.0%)	0.738
**Dyslipidemia**	7 (21.9%)	55 (32.7%)	0.223	3 (14.3%)	0.080	6 (30.0%)	0.919
**CVA**	3 (9.4%)	7 (4.2%)	0.202	2 (9.5%)	0.283	2 (10.0%)	0.263
**PAD**	2 (6.2%)	14 (8.3%)	1.000	2 (9.5%)	0.678	2 (10.0%)	0.665
**Clinical finding**							
**Resting HR**	79.7 ± 11.5	77.0 ± 14.2	0.299	79.3 ± 11.5	0.497	78.5 ± 11.8	0.719
**Resting SBP**	127.8 ± 18.2	124.5 ± 19.0	0.379	131.3 ± 18.5	0.110	126.0 ± 17.5	0.814
**ECG finding**							
**QRS duration**	95.5 ± 21.3	95.3 ± 15.8	0.953	97.4 ± 25.8	0.551	93.8 ± 13.8	0.673
**QTc**	422.6 ± 34.6	424.3 ± 27.8	0.754	426.5 ± 35.2	0.683	425.9 ± 38.1	0.762
**VPC at rest**	7 (21.9%)	3 (1.8%)	0.000	6 (28.6%)	0.000	5 (25.0%)	0.001
**Exercise test**							
**Maximal HR**	157.0 ± 23.2	156.8 ± 25.1	0.954	154.1 ± 20.7	0.604	158.1 ± 24.7	0.805
**Maximal SBP**	190.1 ± 32.4	180.3 ± 27.3	0.074	190.8 ± 35.0	0.130	185.9 ± 33.5	0.507
**Exercise capacity**	11.3 ± 2.3	11.6 ± 2.1	0.468	11.4 ± 2.0	0.809	11.2 ± 2.7	0.402
**TMT positive**	5 (15.6%)	15 (8.9%)	0.329	5 (23.8%)	0.042	3 (15.0%)	0.430

Data are presented as mean ± standard deviation or *n* (%). Abbreviations: BMI, body mass index; FHx, family history; CAD, coronary artery disease; SCD, sudden cardiac death, DM, diabetes mellitus; HTN, hypertension; CVA, cerebro-vascular accident; PAD, peripheral artery disease; HR, heart rate; SBP, systolic blood pressure; VPC, ventricular prem-ature contracture; TMT, treadmill.

**Table 2 medicina-58-00556-t002:** Risk for TMT positive in exercise provoked VPC group (*n* = 153, age ≤ 65).

	TMT Positive Result	
	Univariable	Multivariable
Age	1.105 (1.030–1.186)	1.118 (1.036–1.207)
Sex	0.182 (0.023–1.426)	
BMI	1.012 (0.855–1.199)	
DM	1.892 (0.483–7.410)	
HTN	2.571 (0.901–7.343)	
Dyslipidemia	2.538 (0.890–7.237)	
CVA	6.429 (0.989–41.790)	
PAD	1.248 (0.144–10.843)	
Exercise phase VPC	4.778 (1.279–17.849)	6.879 (1.458–32.453)
Recovery phase VPC	1.500 (0.304–7.396)	

Data are presented as odds ratios (95% confidence interval).

**Table 3 medicina-58-00556-t003:** Risk for CAD and revascularization in exercise provoked VPC group (*n* = 200).

	Coronary Artery Disease		Revascularization	
	Univariable	Multivariable	Univariable	Multivariable
Age	1.095 (1.028–1.165)	1.108 (1.026–1.196)	1.083 (0.984–1.192)	
Sex	0.000 (0.000–0.000)		0.000 (0.000–0.000)	
BMI	1.045 (0.857–1.275)		0.967 (0.697–1.342)	
TMT positive result	1.930 (0.566–6.581)	7.547 (1.812–31.434)	2.267 (0.312–16.472)	8.797 (1.073–72.127)
Exercise phase VPC	1.988 (0.400–9.886)		2.933 (0.291–29.550)	
Recovery phase VPC	2.111 (0.423–10.533)		9.889 (1.313–74.473)	9.381 (1.144–76.948)

Data are presented as odds ratios (95% confidence interval). Included variable: age, sex, BMI, TMT positive result, exercise phase VPC, recovery phase VPC.

## Data Availability

Data sharing not applicable.

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
