# Peer review of "Clinical Significance of Ventricular Premature Contraction Provoked by the Treadmill Test"

_medicina, 2022, doi:10.3390/medicina58040556_

Round 1

Reviewer 1 Report

This is very well written paper and correctly performed study. However, there are some concerns that Authors should address.

Major comments:

  1. Contributions should be clearly stated in the Introduction, especially as this is not much novel approach, although presented data and study design are useful for medical and scientific community. I detected some contributions in later chapters (see for example: "there is paucity of data in Asian populations" in page 6).
  2. Medical background/causes for the presented finding are missing in the Discussion section.

Minor comments:

  1. In Table 1, please, place a mark or highlight results with statistical significance.
  2. Sentence "These findings indicate that the relation between VPCR and revascularization was not due ..." should be placed in the Discussion.
  3. The references are lacking for the paragraph on page 6, lines 153-157
  4. The references are lacking for the paragraph on page 7, lines 191-199
  5. I detected some minor typos (e.g., "results[8, 11]" should be "results [8, 11]") and failed page brakes (see Table 1). Please, correct these.
  6. Please, note that units are missing throughout the manuscript (see for example heart rate in Table 1).

Reviewer 2 Report

Minjung Bak et co-authors presented a very interesting study about treadmil test induced ventricular premature complexes and the resulting clinical significance. 

Treadmil ECG test is rarely used nowadays for ischaemia non-invasive testing. It has poor positive and negative predictive results for functional ischaemia testing. It is a much better test to induce arrhythmias. VPBs are easily recognised phenomena. I agree with distinguising during exercise and during recovery phase VPBs. 

But I consider the clinical significance part inappropriate. Since Treadmil test is very poor to detect ischamia, I would totally delete these results. In my opinion the only interesting clinical outcome is positive coronary angiography or mortality. Even positive CT angiogryphy is not relevant since CT angiography has a very high negative predictive value, but a lower positive predictive value. 

Can the whole clinical significance part be rewritten to just show positive coronary angiography/ PCI / CABG and mortality? 

Round 2

Reviewer 2 Report

I accept the changes. My guess was that VPBs during exercise would be more important but your data suggests that recovery phase VPBs are more prognostic for CV disease. 

Exercise VPBs are important in case of arrhytmogenic diseases. 

I recommend the paper for publication.